# Efficient Flexible All-Solid Supercapacitors with Direct Sputter-Grown Needle-Like Mn/MnO*_x_*@Graphite-Foil Electrodes and PPC-Embedded Ionic Electrolytes

**DOI:** 10.3390/nano10091768

**Published:** 2020-09-07

**Authors:** Apurba Ray, Delale Korkut, Bilge Saruhan

**Affiliations:** 1Department of High-Temperature and Functional Coatings, Institute of Materials Research, German Aerospace Center (DLR), 51147 Cologne, Germany; apurba.ray@dlr.de (A.R.); delale.korkut@dlr.de (D.K.); 2Department of Chemistry, RWTH Aachen University, 52062 Aachen, Germany

**Keywords:** direct-growth, flexible, ionic liquid, Mn/MnO*_x_*, sputter coating, polymer gel electrolyte, supercapacitor

## Abstract

Recent critical issues regarding next-generation energy storage systems concern the cost-effective production of lightweight, safe and flexible supercapacitors yielding high performances, such as high energy and power densities as well as a long cycle life. Thus, current research efforts are concentrated on the development of high-performance advance electrode materials with high capacitance and excellent stability and solid electrolytes that confer flexibility and safety features. In this work, emphasis is placed on the binder-free, needle-like nanostructured Mn/MnO*_x_* layers grown onto graphite-foil deposited by reactive sputtering technique and to the polymer gel embedded ionic electrolytes, which are to be employed as new flexible pseudocapacitive supercapacitor components. Microstructural, morphological and compositional analysis of the layers has been investigated by X-ray diffractometer (XRD), Field Emission Scanning Electron Microscope (FE–SEM) and X-ray photoelectron spectroscopy (XPS). A flexible lightweight symmetric pouch-cell solid-state supercapacitor device is fabricated by sandwiching a PPC-embedded ionic liquid ethyl-methylimidazolium bis (trifluoromethylsulfonyl) imide (EMIM)(TFSI) polymer gel electrolyte (PGE) between two Mn/MnO*_x_*@Graphite-foil electrodes and tested to exhibit promising supercapacitive behaviour with a wide stable electrochemical potential window (up to 2.2 V) and long-cycle stability. This pouch-cell supercapacitor device offers a maximum areal capacitance of 11.71 mF/cm^2^@ 0.03 mA/cm^2^ with maximum areal energy density (E_a_) of 7.87 mWh/cm^2^ and areal power density (P_a_) of 1099.64 mW/cm^2^, as well as low resistance, flexibility and good cycling stability. This supercapacitor device is also environmentally safe and could be operated under a relatively wide potential window without significant degradation of capacitance performance compared to other reported values. Overall, these rationally designed flexible symmetric all-solid-state supercapacitors signify a new promising and emerging candidate for component integrated storage of renewable energy harvested current.

## 1. Introduction

The growing demand for modern flexible portable electronics increases the need for lightweight, low-cost, high-performance compact power sources [1,2,3]. In terms of decentralized energy storage, peak power management, back-up energy storage, the electrochemical energy storage devices, including Li-ion batteries and supercapacitors (SCs) are recognized currently as the most promising devices [4,5]. However, these devices are generally too stiff and bulky to combine into flexible electronics. Moreover, the performances of batteries (Li-ion) are too limited to meet the requirements of harvested energy storage due to their short cycle-life (few hundreds), reaction kinetic, slow rate of mass diffusion and great safety issues [6,7,8]. Hence, the development of low-cost, high performance, flexible supercapacitor offers one of the best solutions to address the above problems. The supercapacitor can bridge the gap between conventional electrostatic capacitors and rechargeable batteries [9,10]. The commercially available supercapacitors (SCs) can provide a reasonably high capacitance value, fast charge–discharge rate (~0.3 to 30 s), long cycling stability (~10^5^) and high energy density and power density compared to conventional capacitor and batteries [11,12]. However, these are bulky and rigid. According to the charge storage mechanism, SCs can be classified into three categories: (1) electric double layer capacitors (EDLCs), (2) pseudocapacitors and (3) hybrid capacitors. In EDLCs, charges are stored via non-Faradic means by formation of electric double layer (EDL) at the electrode/electrolyte interface. No redox reaction or charge transition process occurs in this case [13,14,15]. Usually, highly porous and specific-surface-area-based carbonaceous materials such as carbon nanotubes (CNTs), activated carbon (AC), carbon aerogel, and graphene are used as electrode materials due to their high electrical conductivity [16,17]. In pseudocapacitor SCs, charges are stored via reversible redox reactions on the surface of the electrode. Metal oxides, conducting polymers, are used as pseudocapacitor electrodes. The hybrid supercapacitors combine these charge storage mechanisms (i.e., EDLC and redox reaction) [2,18]. However, the low energy density (~5 Wh/kg) of SCs compared to that of Li-ion batteries (~200 Wh/kg) and relatively high production cost of supercapacitors restrict its wider practical applications. Thus, it is essential to enhance the energy density and long cycling stability of a supercapacitor, by developing advance electrode materials and using suitable synthesis methods. This need has recently attracted immense attention for new and innovative research. The nature of an electrode material and its combination with an appropriate electrolyte play important roles in enhancing the electrochemical charge storage properties of SCs. Electrode materials with high specific-surface-area morphologies result in an enormous increase in the charge storage capacity of an SC device, providing good electrical conductivity, highly accessible electrochemical active sites for electrolyte ions and a short charge-diffusion path length [19,20]. Therefore, the nanostructured transition metal oxides (TMOs) such as RuO_2_, NiO, MnO_2_, V_2_O_5_, Fe_2_O_3_, Co_3_O_4_ etc., having multi-oxidation states, high specific-surface-area and fast redox reaction capability, draw attention in the field of SC electrode material development [12,21,22]. Amongst TMOs, ruthenium oxide (RuO_2_) is a promising electrode material for SC devices relying on excellent pseudocapacitive properties, reversible oxidation states, high specific capacitance value (860 F/g for tubular RuO_2_) and good electrical conductivity, but other characteristics such as high cost, non-abundancy, and toxicity limit its commercial application as an SC material. Hence, there is a focus on alternative TMOs electrode materials, especially abundant manganese oxide (MnO*_x_*) and its composites, due to its eco-friendliness, low cost and high charge storage capabilities, but the increase in the stable potential window to enhance the energy density of the SC is still under debate. Recent studies with MnO*_x_* report relatively low potential windows: The MnO*_x_*/Au-based, all-solid-state microsupercapacitor of Si et al. [23] shows a potential window of only 1.0 V, while Tian et al. prepared a carbon fiber paper-supported Co/Ni-layered double hydroxide MnO*_x_* electrode by a simple two-step electrochemical deposition process for asymmetric supercapacitor (ASC) application exhibiting a potential window of 1.6 V [24]. Reduced graphene Oxide (RGO)-mediated synthesis of Mn_3_O_4_ electrode for ASC by Gao et al. also results in a potential window of 1.6 V [25]. Rafique et al. reported a manganese oxide-based asymmetric supercapacitor displaying stable potential window up to 1.8 V [26]. Saha et al. also fabricated an asymmetric supercapacitor with MnO_2_ microfiber electrodes which were synthesized by Rotary-Jet Spin method and obtained a potential window of 1.6 V [27]. Therefore, it is crucial to design advance flexible binder-free electrodes to improve the stable potential window as well as enhancing the overall conductivity of the active electrode materials [19,25,28].

Thus, this present work deals with its synthesis by sputtering technique, through direct-growth of Mn/MnO*_x_*@Graphite-foil, to yield flexible electrodes. The microstructure and the compositional analysis are carried out with XRD, FE-SEM and XPS measurements. Next, flexible symmetric solid-state pouch-cell supercapacitor cells (SCs) were designed in 3 × 2 cm^2^ single cells by sandwiching ethyl-methylimidazolium bis(trifluoromethylsulfonyl) imide [EMIM][TFSI]: propylene carbonate (PPC) gel electrolyte between two Mn/MnO*_x_*@Graphite foil electrodes. The detailed electrochemical performance of this device was studied by cyclic-voltammetry (CV), galvanostatic charge–discharge (GCD) and electrochemical impedance spectroscopy (EIS) measurements.

## 2. Materials and Methods 

### 2.1. Materials

The ethyl-methylimidazolium bis (trifluoromethylsulfonyl) imide [EMIM][TFSI] (IoLiTec, Heilbronn, Germany), polypropylene carbonate (PPC) from Sigma Aldrich (Taufkirchen, Germany) acetonitrile from Merck, PEDOT:PSS with the brand name Clevious™ from Hereaus Deutschland GmbH and Co. KG (Hanau, Germany) and commercial thin graphite foils of ca. 75-µm thickness purchased from Distrelec (Bremen, Germany). All chemicals were of analytical grade and used without any further purification.

### 2.2. Preparation of the Mn/MnO_x_@Graphite-Foils Electrodes

Commercial thin graphite foils (thickness < 200 μm) were used as substrates for the deposition of coatings. A two-source sputtering equipment from the SVS–Vacuum coating Technologies–SYSTEC Group (Karlstadt, Germany) was used for deposition of Mn/MnO*_x_* layers. A manganese metal target of ~90 mm was employed for sputtering process. After initial deposition of metallic Mn layer by applying 200-W high-frequency (HF) power for approximately 20 min to the metal target, oxygen gas is released into the chamber to enable the deposition of MnO*_x_* layer. This coating sequence is applied in order to provide a sufficiently strong bonding of the coating on the substrate, because manganese oxide sputtering requires low power and thus, yields a low energetic flux. The reactive sputtering process was carried out under the same power and continuous gas flow of oxygen (5 mL/min) + Argon (23 mL/min) for a duration of 5 h to reach an oxide coating layer of approx. 1.5 µm. Prior to the coating deposition, the graphite substrate was etched for 5 min with pure Argon and 400 V BIAS potential applied to the sample. 

### 2.3. Fabrication of Mn/MnO_x_@Graphite-Foil Electrodes Based Flexible All-Solid-State Supercapacitors

For the preparation of the polymer gel electrolyte (PGE), PPC was first dissolved in acetonitrile in a ratio of 1:5 under magnetic stirring at 90 °C for 30 min until the PPC was completely dissolved. Next, [EMIM][TFSI] was added to the polymer matrix in the 1:1 weight ratio [EMIM][TFSI]:PPC. After the addition of the [EMIM][TFSI] ionic salt, the polymer solution was stirred for another 30 min at 90 °C until everything was homogenized. In extension of another 2 h of stirring, gel electrolyte became viscous. Then, a symmetric pouch-cell supercapacitor was assembled in 3 × 2 cm^2^ single cells by sandwiching a thin layer of this ionic liquid:PPC gel electrolytes between two Mn/MnO*_x_*@Graphite-foil electrodes, and it was finally vacuum packaged for further characterizations.

### 2.4. Characterizations

The structural evaluation of the thin oxide films was done with a XRD diffractometer D-5000 equipment from SIEMENS (Erlangen, Germany) with Cu K_α_ radiation (λ = 1.54178 Å). The EVA software from BRUKER AXS (Karlsruhe, Germany) and the JCPDS database were employed to assign the reflections to the experimental patterns. The microstructure and the compositional analysis were carried out with a Field Emission Scanning Electron Microscope (FE-SEM) Zeiss ULTRA 55™ (Jena, Germany) equipped with an Energy Dispersive X-Ray Spectrometer (EDS) from Oxford Instruments (Wiesbaden, Germany). The XPS measurement was carried out in Cinvestav Querétaro in México using a monochromatic XPS-equipment, Alpha 110, ThermoFisher Scientific (Apodaca, México). 

The electrochemical characterizations of SC devices based on as-fabricated Mn/MnO*_x_*@Graphite-foil electrodes was studied using an electrochemical workstation, Reference 3000 equipped with EIS 300 from Gamry Instruments (C3 Prozess-und Analysentechnik, GmbH, Haar near Munich, Germany) using CV, GCD and EIS tests. The electrochemical impedance spectroscopy (EIS) data were recorded in the frequency range between 0.01 Hz and 100 kHz with an AC magnitude of 10 mV, and fitted with EC lab software to get the equivalent circuit model. The areal capacitance (*C_A_*) (F/cm^2^) of the device can be calculated from CV curve using Equation (1) [29,30]
(1)$$C_{A}\left(F/{\mathrm{cm}}^{2}\right)=\;\frac{1}{2\times A\times \nu \times \left(V_{f}-V_{i}\right)}\int_{V_{i}}^{V_{f}}I\left(V\right)dV$$
where:
*A*: Effective area of a single electrode or effective area of the device, (cm^2^);*ν*: scan rate, (mV/s);(*V_f_* − *V_i_*): potential window (∆*V*), (*V*);$\int_{V_{i}}^{V_{f}}I\left(V\right)dV$: area under the CV curve.


Areal capacitance (*C_A_*), areal energy density (*E_a_*, mWh/cm^2^) and areal power density (*P_a_*, mW/cm^2^) can be calculated from discharge curve of GCD using the following Equations (2)–(4)
(2)$$C_{A}\left(F/{\mathrm{cm}}^{2}\right)=\;\frac{I\times ∆t}{A\times ∆V}$$
(3)$$E_{a}\left(\mathrm{mWh}/{\mathrm{cm}}^{2}\right)=\;\frac{C_{A}\times ∆V^{2}}{7.2}$$
(4)$$P_{a}\left(\mathrm{mW}/{\mathrm{cm}}^{2}\right)=\;\frac{E_{a}\times 3600}{∆t}$$
where:
*I*: discharge current, (A);Δ*t*: discharge time, (s);


## 3. Results and Discussion

### 3.1. Structural and Morphological Analysis

The phase analysis of sputter-grown Mn/MnO_*x*_@Graphite-foil electrodes and bare graphite foil was performed by X-ray diffraction (XRD) analysis. The XRD analysis (Figure 1a) exhibits that in the as-coated case, Mn/MnO*_x_* coatings deposited on graphite foils are amorphous. The amorphous nature of the oxide films was also confirmed by low-angle XRD measurements of the films deposited on quartz substrates. The XRD pattern clearly indicates two highly intense peaks at 2θ values of 25.2° and 55.4°, which are mainly belong to graphite substrate corresponding to the diffraction planes (002) and (101) (JCPDS card no. 00-001-0640) [31]. The XRD patterns of bare graphite substrate and after Mn/MnO*_x_*-coated graphite substrate seem similar and it is very difficult to distinguish the Mn/MnO*_x_* peaks from this XRD pattern due to their very low intensities compared to graphite substrate. Only after heat-treatment at 450 °C, can a low-intensity broad peak around 2θ values of 35–80° (Figure 1a (inset)) composed of more peaks be observed, which is mainly due to the formation of a weekly crystalline Mn_2_O_3_ phase(JCPDS card no. 00-41-1442) on the surface of graphite substrate [18,32]. The small shift in graphite peaks towards lower diffraction angle and decrease of intensities have been observed after heat-treatment Mn/MnO*_x_* coating, which also signified the formation of MnO*_x_* on the surface of graphite foil.

The surface morphology (Figure 1b) shows that the whole surface of graphite foil substrate is uniformly covered by porous needle-like Mn/MnO*_x_* nanostructures. This sputter-grown, needle-like morphology of Mn/MnO*_x_* nanostructures onto graphite foil substrate provides a sufficient electrochemically active, high surface area to the electrolyte ions [28,30,33]. The FESEM images revealed the interlinked dendritic needle-like nanostructures not only improve the electrical conductivity through electron transport and electrolyte ions diffusion but also offer enough sites for Faradaic redox reactions through Mn/MnO*_x_*. High magnification image of this electrode (see Figure 1b (inset)) clearly exhibits the equalized grains growth reaching to as large as 50 nm diameter and 200–300 nm length, that considerably increases the accessibility of the maximum redox active sites and reduces the effective charge transfer resistance, leading to the enhancement of capacitance value.

The oxidation states of this sputter-grown Mn/MnO*_x_*@Graphite-foil electrode were studied via XPS analysis. The XPS spectra of Mn 2p orbit (Figure 1c) shows two prominent peaks at binding energy (BE) of 640.75 and 652.39 eV, corresponding to Mn 2p_3/2_ and Mn 2p_1/2_ states. The BE of Mn in the pure oxide (Figure 1c) is 640.7 eV, which also matches the oxidation state close to Mn_2_O_3_ or Mn (III). The separation value between these two states (Mn 2p_3/2_, Mn 2p_1/2_) of Mn is 11.64 eV, which is also consistent with the literature [4,34,35]. The BEs of 528.12 and 529.46 eV from oxygen indicate that oxygen is bonded to the metal and only a few hydroxyl groups are present (Figure 1d). These few OH groups present in our coatings can come from the exposure of the electrodes to and their reaction with the atmospheric moisture after opening the sputtering coating chamber. The BE of 528.12 eV of O 1s (Figure 1d) is mainly due to structural oxygen in MnO*_x_*. No information regarding the presence of oxygen coming from hydroxides is observed. Structural oxygen in the bond Mn–O–Mn is found around 529.46 eV, and this signal depends on the XPS equipment conditions and sample history [18,36]. Hence, from this comprehensive XPS analysis, it can be concluded that the Mn/MnO*_x_* film has been successfully coated on graphite-foil substrate.

### 3.2. Electrochemical Properties of Flexible Supercapacitor Based on Mn/MnO_x_@Graphite-Foil Electrodes

For practical applications, the electrochemical study of a supercapacitor cell using two-electrode measurements are more reliable to investigate the electrochemical properties of the electrodes, including the synthesis method, than those of three electrode measurements [2,37]. The electrochemical performance of this flexible supercapacitor (FSC) cell based on Mn/MnO*_x_*@Graphite foil electrodes and polymer gel electrolyte is given in Figure 2. A series of CV measurements (Figure 2a) of this symmetric pouch-cell supercapacitor device were performed by applying different potential windows varying from 0–1.4 to 0–2.4 V at 80 mV/s to estimate the maximum stable operating potential window of this FSC. It is well-known that a higher stable potential window extends the desired output as an alternative to the series combination of devices [25,38]. In this present case, no obvious increase in positive current (anodic current) due to any polarization has been observed even at 2.2 V for this SC, which implies that this SC device can be performed up to a wide stable potential window of 0–2.2 V. Therefore, this potential window (e.g., 0–2.2 V) is chosen for safe and stable results at further electrochemical investigations of the FSC with a polymer gel electrolyte in 1:1 ratio of [EMIM][TFSI]:PPC. Figure 2b shows this all-solid-state FSC measured at different scan rates from 5 to 100 mV/s within the chosen potential window of 0–2.2 V yields almost rectangular CV curves at each scan rate, suggesting its excellent rate capability and reversibility. The maximum capacitance contribution from pseudocapacitance can be confirmed by the presence of broad redox peaks in this potential window of up to 2.2 V at each scan rate. The increase in capacitive current with increasing scan rate from 5 to 100 mV/s is also observed, implying the promising capacitive behaviour of this FSC. The needle-like structures of the Mn/MnO*_x_* electrodes provide a high surface area with a high number of nanopores for the maximum diffusion of electrolyte ions from [EMIM][TFSI]: PPC polymer gel electrolyte at the electrode/electrolyte interface, leading to high capacitance values. The scan-rate-dependent areal capacitance (C_A_) of this device was calculated from CV curves using Equation (1).

A maximum areal capacitance of 8.69 mF/cm^2^ is obtained at 5 mV/s for this SC. The variation in C_A_ with scan rate (Figure 2c) shows that areal capacitance values decrease significantly with increasing the scan rate due to decrease of time involvement for the movement of electrolyte ions [4,39,40]. To further evaluate the areal capacitance (C_A_) of this all-solid FSC device, the galvanostatic charge–discharge (GCD) tests at different current densities from 0.03 to 1.0 mA/cm^2^ were carried out. The GCD curves in Figure 3a show the obtainment of nearly linear and symmetric shape at different curve densities, suggesting almost ideal capacitive behaviour from this FSC. The areal capacitance of this device was evaluated from the discharge curve at different current densities using Equation (2). The variation in C_A_ with current densities (Figure 3b) represents that C_A_ value decreases with increase in current density due to the time limitation of electrolyte ions’ movement within the electrode pores [41,42]. The maximum C_A_ 11.71 mF/cm^2^ at 0.03 mA/cm^2^ was achieved, which is almost equal to the value calculated from CV curve at 5 mV/s. The Ragone plot (Figure 3c) calculated from GCD curves using Equations (3) and (4) of this FSC exhibits the highest areal energy density (E_a_) of 7.87 mWh/cm^2^ at an areal power density (P_a_) of 36.65 mW/cm^2^ and still maintains 2.84 mWh/cm^2^ at a higher power density of 1099.64 mW/cm^2^. To our best knowledge, there is no other higher reported value of energy and power densities obtained for a MnO*_x_*-based SCs with this wide a stable potential window of up to 2.2 V than the one we now report, with this all-solid flexible SC having the direct sputter-grown needle-like Mn/MnO*_x_*@Graphite-foil electrode and polymer gel electrolyte (see Appendix A). Moreover, relying on the over 5000 cycles that were performed with CV measurements at a scan rate of 100 mV/s (Figure 3d), it is foreseeable that this symmetric SC cell of Mn/MnO*_x_*@Graphite-electrode and ionic-liquid-embedded polymer gel electrolyte promise long-term cycling stability. The CV curves remain almost the same for all throughout these cycles. The slight shift in the CV curves and the insignificant loss of capacitance may be due to electrochemical dissolution of the Mn/MnO*_x_* electrode materials.

This almost-constant remaining nature of the CV curves indicates the achievement of good cycling stability with the FSC device obtained with this polymer gel electrolyte and Mn/MnO*_x_*@ Graphite-foil electrodes.

The EIS studies that were performed in the form of Nyquist plots (Figure 4a) before and after 5000-cycle CV measurement at 100 mV/s to further investigate the capacitive, resistive, charge-transport properties confirm the long-term cycling stability of this FSC device. These two EIS curves show almost the same patterns consisting of a small semicircle in a high-frequency region, corresponding to charge transfer resistance (*R_ct_*), and a straight line portion in low-frequency region, that corresponds to the ion diffusion controlled Warburg impedance (*W*) at the electrode/electrolyte interface [43,44]. The intersect point of the semicircle in Z′-axis at high frequency represents the contact resistance of the electrode/electrolyte interface (*R_s_*) and electrolyte resistance. An equivalent electrical circuit (Figure 4a (inset)) was modeled using EC lab software, indicating that the circuit consists of three distinct parts. The first part gives the value of *R_s_*, second and third gives the value of *R_ct_* and constant phase elements (CPE), which is included due to the inhomogeneity, nonlinearity and variation in potential in the system. The parallel combination of CPE1 and *R_ct1_* in the second portion arises due to kinetics of the Mn/MnO*_x_*@Graphite electrode, and the last portion (CPE2 II *R_ct2_*) signifies the kinetics of the electrode/electrolyte interface. Figure 4a shows that the *R_s_* value and *R_ct2_* decrease from 11.7 to 10.96 Ω and 25 to 21.97 Ω, respectively, due to activation of the electrode materials during 5000 charge/discharge cycles. The slight shifting of the straight-line portion in the low-frequency region observed after 5000 CV, is mainly due to an increase in the repulsive resistive effect by already-occupied ions in the pores. The goodness of fit (χ^2^) for this device was calculated as 0.02 and 0.05, respectively, suggesting a good fitting. Hence, the relatively low *R_s_* and *R_ct_* values are advantageous for the high capacitance and long cycle life of this Mn/MnO*_x_*@Graphite electrode SC. The Bode plots given in Figure 4b before and after 5000 cycles CV of this symmetric Mn/MnO*_x_*@Graphite SC cell show a maximum phase angle at low frequency (0.01 Hz) of 61.58° and 54.35°, respectively. In general, the maximum phase angle at low-frequency approaches, 90°, reveals a pure capacitive nature for a supercapacitor device [45,46]. The phase angle for this SC device suggests the promising capacitive behavior of this SC. The shift in phase angle from ideal capacitor value (90°) for this SC is mostly due to the pseudocapacitive nature of the Mn/MnO*_x_* electrode materials. However, a broad peak was observed for both cases at high frequency due to the diffusive resistance of this symmetric SC. The peak frequency values of this SC before and after 5000 CV are 12.5 and 7.2 kHz, respectively, which suggests relatively low diffusion resistance for the electrolyte ions of this device [47]. To better understand the intrinsic charge conduction properties of the as-prepared electrodes, the evolution of the complex capacitance has been performed. In general, the SCs can be described using a series combination of resistance (*R*) and capacitance (*C*) that depends upon the pulsation (10 mV in this case) *ω* (*ω* is the angular frequency 2πf) The frequency dependence capacitance *C*(ω) can be expressed as a combination of the real *C*′(*ω*) and imaginary C″(*ω*) parts of capacitance, which are written as Equations (5)–(8) [45,46].
(5)$$C\;\left(\omega \right)=\;\frac{-1}{\omega Z^{\prime \prime }}$$
thus,
C(𝜔) = C′(𝜔) − jC″(𝜔)
(6)
(where j = √-1).

Then, C′(𝜔) and C″(𝜔) can be obtained in the form of impedance as follows
(7)$$C^{\prime }\left(\omega \right)=\;\frac{Z^{\prime \prime }\left(\omega \right)}{\omega │Z\left(\omega \right){│}^{2}}$$
(8)$$C^{\prime \prime }\left(\omega \right)=\frac{Z^{\prime }\left(\omega \right)}{\omega │Z\left(\omega \right){│}^{2}}$$

Here, |Z(𝜔)| is the modulus of impedance. The variation in C′(𝜔) with applied frequencies for this SC cell built of Mn/MnO*_x_*@Graphite-foil electrodes and PGE before and after 5000 cycles CV (Figure 4c) shows a resistive nature at high-frequency and a promising capacitive nature at low-frequency regions. The low-frequency (at 0.01 Hz) capacitance values for this symmetric SC decrease from 4.34 to 3.99 mF/cm^2^ after 5000 cycles CV, which also propose the good cycle stability of these SC electrodes. The change in C″(𝜔) with frequencies (Figure 4d) before and after 5000 cycles CV represents a flat peak corresponding to a peak frequency (f_0_), which also explains the capacitive and resistive behavior of this SC cell. The relaxation time (τ_0_) was calculated from τ_0_ = 1/2πf_0_ from the peak frequency for before and after 5000 CV. If the frequency (f) is greater than peak frequency (f_0_), i.e., f > f_0_, then the SC device works as a pure resistor and when f < f_0_, then SC works as a capacitor. The relaxation time (τ_0_) values for this SC cell consisting of Mn/MnO*_x_*@Graphite-foil electrodes and PGE are 0.78 and 0.54 s before and after 5000 CV cycles, respectively. These low values of the relaxation time signify the high-energy storage capability with a high rate and fast reversible adsorption/desorption of electrolyte ions at the electrode/electrolyte interface [4,46,48]. The overall electrochemical properties suggest very promising supercapacitive properties for the FSC cell with the direct sputter-grown needle-like Mn/MnO*_x_*@Graphite-foil electrodes and ionic-liquid-embedded polymer gel electrolyte (PGE) for supercapacitor applications where high performance and all-solid flexible devices especially in the next generation energy storage systems are required.

## 4. Conclusions

A needle-like Mn/MnO*_x_* nanostructures was successfully synthesized onto a flexible graphite-foil substrate by sputter technique. Pouch-cell type and all-solid flexible supercapacitor cells are manufactured by sandwiching Mn/MnO*_x_*@Graphite-foil electrodes with an [EMIM][TFSI]:PPC-based polymer gel electrolyte exhibiting a satisfactory supercapacitive performance in terms of an areal capacitance of 11.53 mF/cm^2^@0.03 mA/cm^2^ with maximum areal energy density (E_a_) of 7.87 mWh/cm^2^ and areal power density (P_a_) of 1099.64 mW/cm^2^, wide potential window of 2.2 V, yielding low resistivity, flexibility and good cycling stability. Therefore, these rationally designed FSCs signify the development of a hopeful new candidate for employment in applications where renewable energy harvested current storage is required.

## Figures and Tables

**Figure 1 nanomaterials-10-01768-f001:**
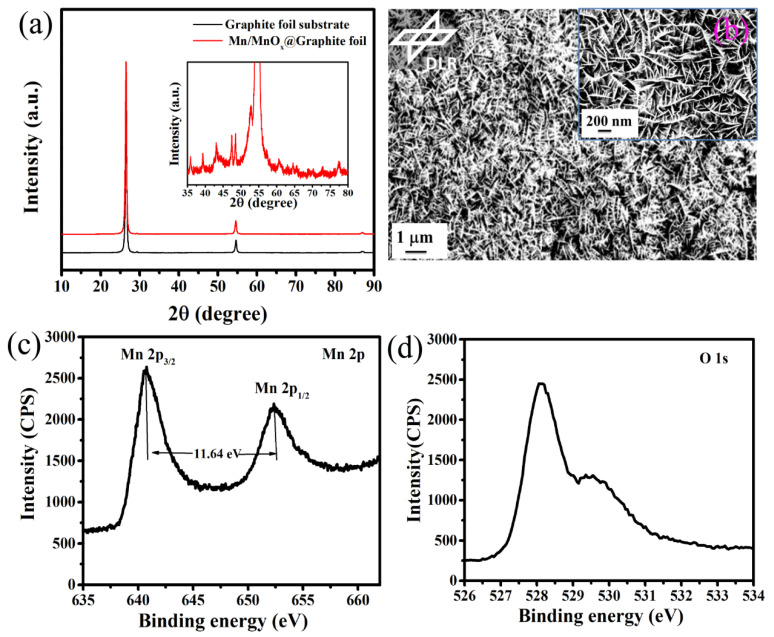
(**a**) XRD patterns, (**b**) FE-SEM images, (**c**,**d**) XPS spectra of Mn 2p and O 1s of Mn/MnO*_x_*@Graphite-foil electrode.

**Figure 2 nanomaterials-10-01768-f002:**
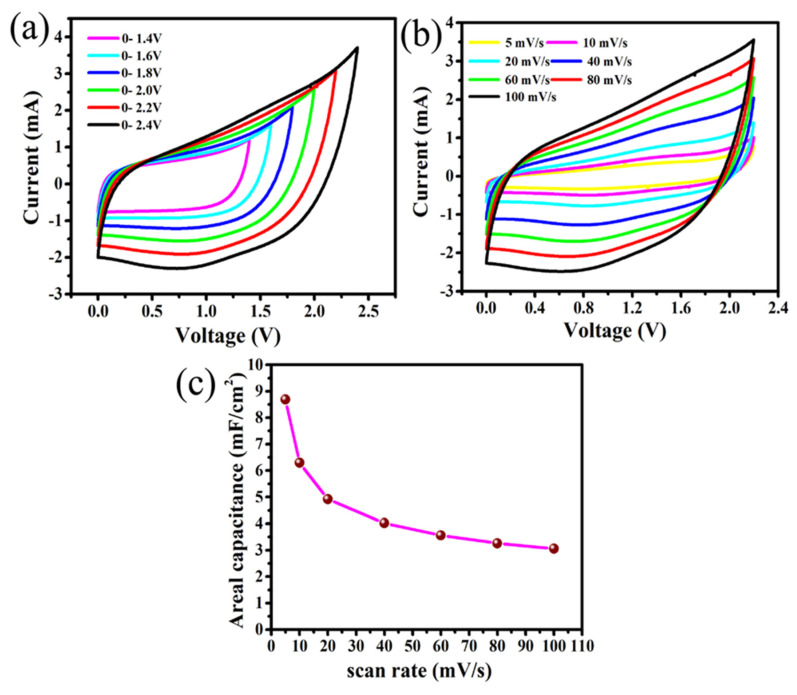
(**a**) Cyclic-voltammetry (CV) at different potential windows, (**b**) CV at different scan rate and (**c**) areal capacitance vs. scan rate of flexible supercapacitor (FSC) based on Mn/MnO*_x_*@Graphite-foil electrodes and polymer gel electrolyte.

**Figure 3 nanomaterials-10-01768-f003:**
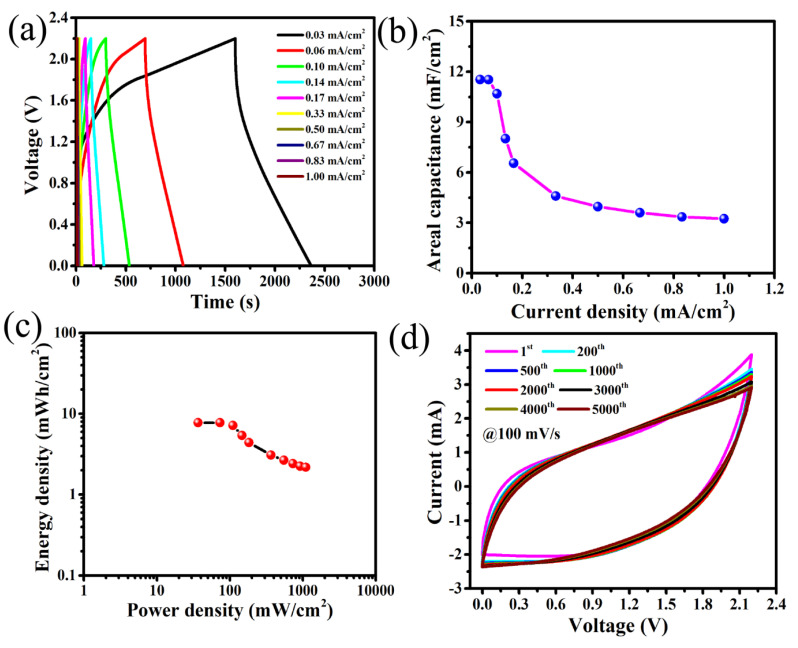
(**a**) Galvanostatic charge–discharge (GCD) at different current densities, (**b**) areal capacitance vs. current density, (**c**) Ragone plot and (**d**) 5000 CV measurements of Mn/MnO*_x_*@Graphite electrodes based FSC.

**Figure 4 nanomaterials-10-01768-f004:**
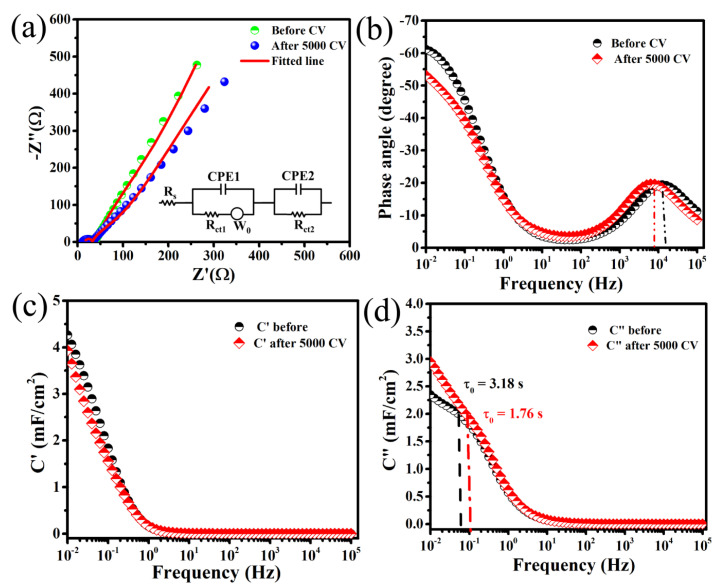
(**a**) Electrochemical imprudence spectroscopy (EIS) before and after 5000 cycles with equivalent circuit (inset), (**b**) Bode plots, (**c**) Real part of capacitance (C′) vs. frequency and (**d**) Imaginary part of capacitance (C″) vs. frequency of FSC based on polymer electrolyte gel and Mn/MnO*_x_*@Graphite-foil electrodes.

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
