# Peer review of "Efficient Flexible All-Solid Supercapacitors with Direct Sputter-Grown Needle-Like Mn/MnOx@Graphite-Foil Electrodes and PPC-Embedded Ionic Electrolytes"

_nanomaterials, 2020, doi:10.3390/nano10091768_

Round 1

Reviewer 1 Report

In this work, the authors describe a novel type of flexible supercapacitor with nanostructured electrodes fabricated by sputtering. The work is interesting and the performance of the supercapacitor is promising. The manuscript could be improved by taking into account the follow ing suggestions:

  • it would be better to replace the term "in-situ synthesis" in the abstract and the text by "deposited by sputtering". In-situ synthesis is a special type of chemical synthesis of nanomaterials, not the case here. where two different layers are deposited by a physical method.
  • Figure 1 is not really necessary, the procedure is sumple and can be explained in the text
  •  Figure 2(d) has to be improved substantially. The sample should be measured at a higher magnification and a scale bar   is necessary.
  • line 220-229 which summarize literature  data would be better included in the introductory part.

Author Response

The The authors are very thankful for the constructive comments of the Reviewer that will definitely enhance the quality of the manuscript.  We have revised the manuscript in accordance with the comments (All the changes are highlighted in yellow in the revised manuscript file). Here we present the replies to the specific comments of the reviewer:

1-The term "in-situ synthesis" is replaced by "deposited by reactive sputtering"in the abstract and the text. Please see yellow highlighted.

2-Figure 1 is removed

3-Figure 2(d) has been improved as suggested and is now Figure 1: higher magnification and scale bars are added.

4- line 220-229 is now moved to line 79-93

Reviewer 2 Report

This manuscript describe the result on synthesis of Needle-like Mn/MnOx@Graphite for supercapacitors. Results are interesting and certainly adds new information in the field of supercapacitors. However, reviewer feel that few corrections are necessary to improve the scientific quality of this paper as suggested below 1. For supercapacitor application, 5000 cycles are not enough as ehen consider for design device which have stable performance. It is recommended that 10000 cycles are enough for supercapacitor measurement. Author can explain this. 2. There are several reports on Mn in combination with CNT, graphite , graphene and so on. Author can prepare table of comparison to show that how the present result are better than earlier published data.

Author Response

The authors are very thankful to the editor and the reviewers for the swift review and insightful comments about the manuscript.  The constructive comments of Reviewers will definitely enhance the quality of the manuscript.  We have revised the manuscript in accordance with the comments (All the changes are highlighted in yellow in the revised manuscript file). Here we present the replies to the specific comments of the reviewers.

Round 2

Reviewer 1 Report

The authors amended the manuscript according the suggestions of the reviewer.